# Experimental Investigation of the Vortex Dynamics in Circular Jet Impinging on Rotating Disk

**Mouhammad El Hassan** [1,*] and **David S. Nobes** [2]

1   Mechanical Engineering Department, Prince Mohammad Bin Fahd University, Al Khobar 34754, Saudi Arabia
2   Department of Mechanical Engineering, University of Alberta, Edmonton, AB T6G 2R3, Canada; david.nobes@ualberta.ca
*   Correspondence: melhassan@pmu.edu.sa

**Abstract:** A circular jet impinging perpendicularly onto a rotating disk is studied in order to understand the influence of centrifugal forces on the radial wall jet. Time-resolved Particle Image Velocimetry (TR-PIV) measurements are conducted in different jet regions in order to investigate the flow physics of the large-scale vortical structures and the boundary layer development on the impinging wall for both stationary and rotating impinging disks. The Reynolds number is $Re_D$ = 2480, the orifice-to-plate distance H = 4D (D is the jet-orifice diameter) and the rotation rate is 200 RPM. It is found that the rotation of the impinging wall results in strong centrifugal effects, which affect different regions of the jet. Both radial velocity profiles and turbulence intensity distributions show different behavior when comparing the stationary and rotating cases. Finite Time Lyapunov Exponent (FTLE) analysis is implemented to describe the time-resolved behavior of the large-scale vortical structures and flow separation.

**Keywords:** impinging jet; rotating disk; vortex dynamics; PIV; FTLE





## 1. Introduction

Impinging jets have been extensively investigated because of their wide industrial applications, ranging from turbine blade cooling, to drying processes and cooling of electronic devices, etc. Despite their geometric simplicity, the flow physics of impinging jets is complex [1–4]. The flow dynamics of a jet impinging on a rotating disk has received less attention than with stationary disks despite its relevance to cooling processes in many industrial applications, such as the cooling of bearings, gas turbines disks and alternators of wind generators [5–7]. The focus of the existing literature on jets impinging on a rotating disk is to investigate the jet flow in a rotor-stator configuration. The main difference in the present jet configuration is the absence of a flow confinement around the jet exit, which would highly affect the flow dynamics.

Different types of impinging jets were studied in the literature, including continuous and synthetic jets. It should be noted that the synthetic jets were widely used in mixing and heat transfer applications [8–10]. Zhang et al. [10] investigated the flow dynamics of a pulsed jet impinging on a rotating wall using CFD simulations. These authors evidenced the existence of a pair of entraining vortices and rotating vortices are formed on both sides of the jet and on the rotating wall. In the present paper, however, a continuous jet is investigated.

An impinging jet is usually decomposed into three regions: the free jet, the impinging region and the outer wall-jet region [1–5]. The physical composition of impinging jets depends upon a number of parameters, such as Reynolds number, orifice shape, orifice-to-plate distance and inflow turbulence. Each region of an impinging jet features different turbulence dynamics and requires an in-depth flow physics analysis using advanced experimental techniques. Numerous studies on heat transfer and fluid flow in jet impinging

on rotating surface illustrated the complexity of the fluid flow for simple geometries [11]. The boundary layer development along the impinging wall and the strong streamline curvature in the impinging region are some of the characteristics, which make the flow very complex [12,13]. Therefore, a deep understanding of the flow physics in such impinging jet configurations would be of benefit and can help defining passive and active flow control methods for specific industrial applications. A review of the most important findings for the jet flow impinging on a rotating disk available in the literature is presented in [14]. This literature review focuses on the jet flow patterns and heat transfer on the impinging plate.

The effect of the rotational speed and the geometry of the impinging plate on the characteristics of the boundary layer that develops on the rotating wall was studied by many authors [15–17]. Manceau et al. [15] studied experimentally and numerically the rotational effect of an impinging disk on the heat transfer and flow dynamics of a circular jet for a Reynolds number of 14,500 and an orifice-to-plate distance of five jet diameters. These authors found that the rotational effect does not directly affect the outer layer, but only the inner layer of the wall jet. They found that the boundary layer thickness $\delta$ increases as the plate radius increases and as the rotational velocity decreases. It was also found that the number of vortices generated due to the jet impact is strongly affected by the local Reynold number and that the centrifugal force leads to the generation of more vortices in different regions of the impinging jet [7].

The primary objective of the present study is to deepen the limited knowledge on the flow dynamics developing in different regions of a circular jet impinging on a rotating disk through detailed experimental measurements, not only in terms of the mean velocity and turbulence statistics distributions, but also from a fundamental understanding of the vortex dynamics and their interaction with the impinging wall. Therefore, the complex flow physics of the impinging jet is investigated using time-resolved particle image velocimetry (TR-PIV) measurements. Advanced post-processing and the vortex identification method, namely "Finite Time Lyapunov Exponent" (FTLE) is implemented in order to deeply investigate the spatio-temporal development of the vortical structures. To the authors' knowledge, such an approach was never proposed in the literature and would therefore be of high interest to the scientific community.

## 2. Experimental Apparatus and Procedure

### 2.1. Jet Flow Facility

The experimental setup shown in Figure 1a consists of a jet nozzle positioned inside a rectangular tank. The impinging wall consists of a circular disk attached to the bottom section of a vertical cylinder that is mounted inside a slewing ring. The water jet impinges upward on the rotating disk and the rotating motion is generated using a programmable electric servomotor connected to the slewing ring using a timing belt and a timing pulley (gear reducer ratio of 4:1). The structure supporting the aquarium tank and the motor is mounted on an optical table to reduce any potential vibrations in the system. In order to achieve low turbulence intensity and a uniform axi-symmetric velocity profile at the jet exit, a converging chamber with a fifth order polynomial shape into the jet nozzle, two honey combs and a settling chamber were used (Figure 1b). The Reynolds number based on the jet exit velocity and the jet diameter, $D = 8$ mm, was $Re_D = 2480$. The rotational speed of the impinging disk was 200 RPM, its diameter is $r_{disk} = 5.8\ D$ and the jet exit is located at $H = 4\ D$ from the impinging wall.

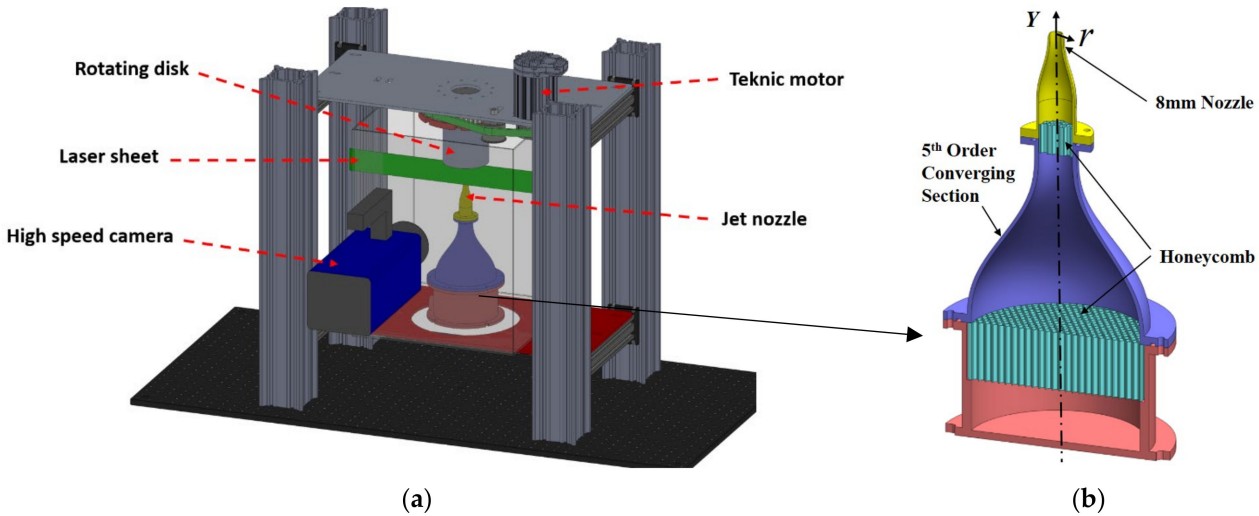

**Figure 1.** Schematic view of (**a**) the experimental setup and (**b**) the jet flow conditioning.

### 2.2. Time-Resolved Particle Image Velocimetry

Velocity fields in the jet and impingement regions are obtained using TR-PIV. The system is composed of a Phantom V611 camera of $1200 \times 800$ pixels and a Photonics Industries Nd:YLF laser of 30 mJ energy and 527 nm wavelength. Small glass spheres, 40 μm in diameter, are used as a tracer for the PIV measurements. Data sets of 2500 PIV image pairs were acquired at a frequency of 500 Hz for each acquisition run. The synchronization between the laser and the camera is controlled by a High-Speed Controller and the data acquisition is performed with DaVis 8.4 software. The images are processed by an adaptive multigrid algorithm correlation handling the distortion window and the sub-pixel window displacement. The final grid is composed of $32 \times 32$ pixel-size interrogation windows with 50% overlap leading to a spatial resolution of 0.6 mm. The maximal displacement error is equal to 1.1% and 1.7% for the longitudinal and the vertical velocity components [18]. The accumulation of the rms error and the bias error gives a total error of about 3.2% of the mean axial velocity.

### 2.3. Finite-Time Lyapunov Exponent (FTLE)

The FTLE, used for vortex identification in the present study, is a measure of the rate of separation of initially adjacent particles [19]. Attracting material or attracting Lagrangian coherent structures (LCS) may be represented by ridges in the FTLE field when the FTLE is calculated by integrating trajectories in backward time $T < 0$ [19]. The FTLE is given by

$$\sigma_{t_0}^T(x) = \frac{1}{T} \, ln \sqrt{\lambda_{max}(\Delta)}$$

where $\lambda_{max}(\Delta)$ is the maximum eigenvalue of the Cauchy-Green deformation tensor, computed for the initial time $t_0$ over an integration time $T$. Mathematically, the LCSs are objective and Galilean invariant, and therefore are independent of any translation or rotation/acceleration of the coordinate frame.

In the present paper, a similar approach to that used in [1] is used to compute the FTLE fields. A backward integration time is chosen to extract the vortical structures in the flow. The computation of trajectories is stopped if they exit the studied domain [20]. The Lagrangian coherent structures are identified by finding local maxima of the FTLE field with the time parameter (non-dimensionalized by $D/U_0$), $T = -10$.

## 3. Results and Discussion

### 3.1. Radial Velocity Profiles

In order to study the influence of the impinging wall rotation on the velocity distribution, mean radial velocity profiles are extracted at different radial locations of the impinging wall for both the stationary and rotating disk, as shown in Figure 2. In the present study, the impinging region corresponds to $r/D < 1.7$ and the near-wall region, $Y/D < 0.1$. It is found that in the impinging region, the radial velocity is smaller with the rotating disk as compared to the stationary case ($r/D = 0$ is the jet centerline) in the near-wall region. However, the radial velocity becomes higher in the outer region of the impinging wall with the rotating disk as compared to the stationary case. Farther from the impinging wall ($Y/D > 0.1$), an opposite trend is observed.

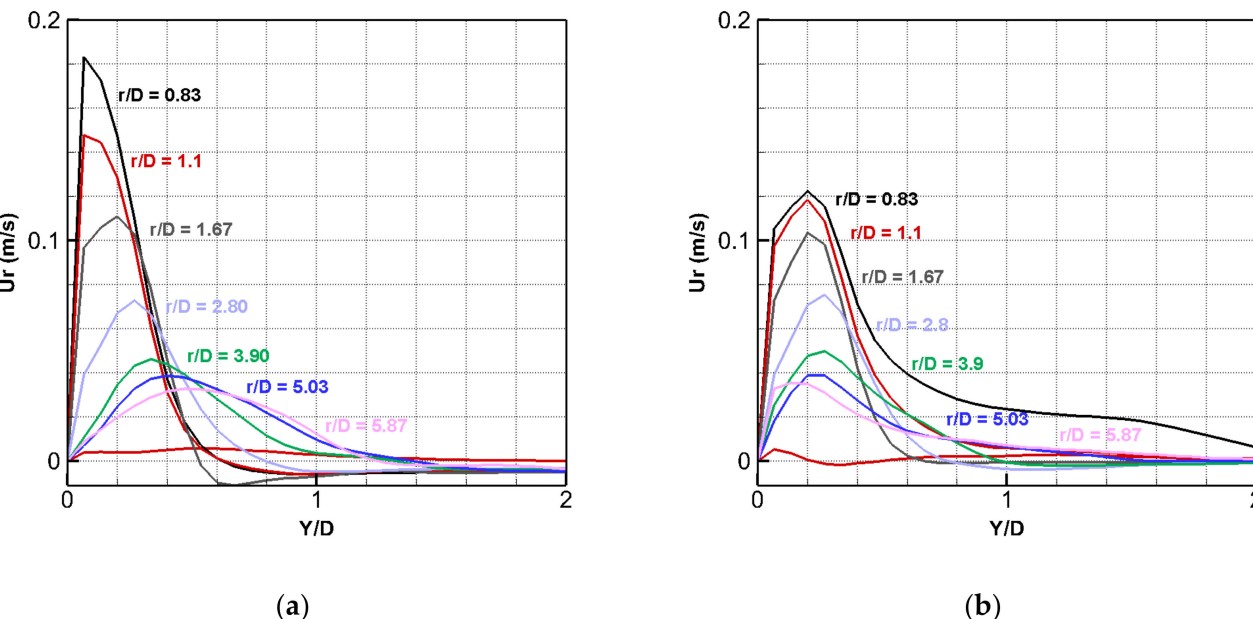

**Figure 2.** Mean radial velocity profiles for the impinging jet onto (**a**) stationary and (**b**) rotational disk (200 RPM).

To the author's knowledge, such distribution of the radial velocity in the impinging region has not been investigated in the literature and it therefore would be of significant interest to better understand the flow physics responsible of such velocity distribution. For a similar jet configuration using a numerical approach, Abdel-Fattah [21] found that the influence of the rotating disk appears only in the outer jet region. The difference between [21] and the present results could be attributed to an inaccurate prediction of the vortex dynamics and flow separation in the impinging region in Abdel-Fattah's study.

Figure 2 also shows that the maximum radial velocity is observed at $r/D = 0.83$. Minagawa and Obi [22] experimentally found a maximum radial velocity around $r/D = 1$. It can be suggested that this difference is due to a greater jet shear layer growth in [22] as compared to the present investigation due to the larger distance between the jet exit and the impinging wall.

### 3.2. Flow Field Statistics

Radial and longitudinal turbulence intensity distributions are plotted in Figure 3 for the stationary and the rotating cases. For the rotational case, the radial turbulence intensity ($Ur,rms$) presents higher values upstream from the impinging plate and in the outer near-wall region of the plate—Regions 1 and 3 in Figure 3a,b. The vertical turbulence intensity ($Uy,rms$) shows higher amplitude in the impinging region—Region 2 in Figure 3c,d, with the stationary disk as compared to the rotational case. This suggests that the centrifugal

force of the rotational disk results in inducing swirl into the free jet just upstream from its impingement on the rotating wall. Such jet swirl results in a modified vortical structures and enhanced mixing which would be responsible for the distribution of the radial RMS observed in Region 1. The centrifugal force also results in an advection of the large vortical structures closer to the impinging wall when traveling from the jet center outward in the radial direction. For the stationary case, the large vortical structures separate from the wall in Region 2. This results in a lower vertical turbulence intensity observed in Region 2 for the rotational case and the higher radial turbulence intensity close to the wall in the outer region—Region 3 in Figure 3a,b.

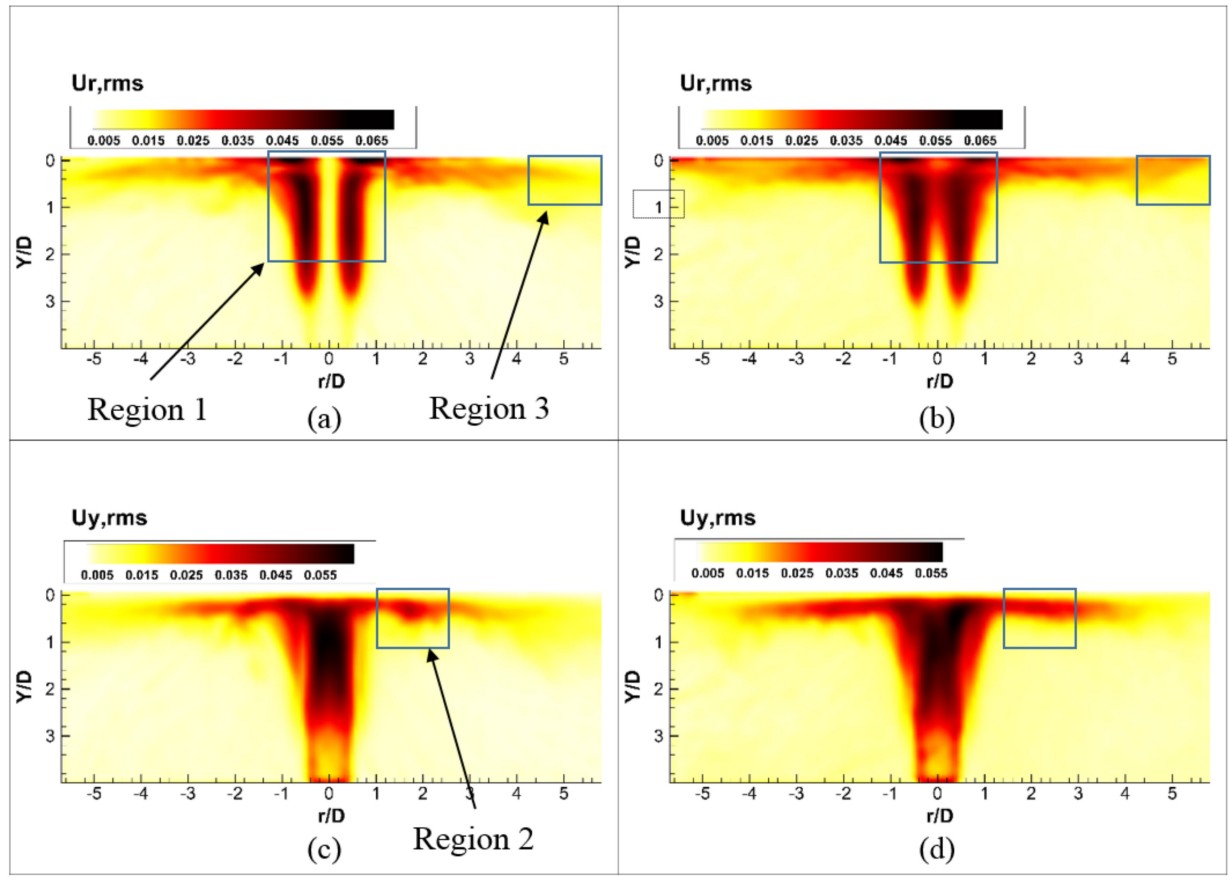

**Figure 3.** Turbulence intensities distribution for (**a**,**c**) stationary and (**b**,**d**) rotating disk.

### 3.3. Instantaneous Velocity Fields and Vortex Dynamics; Eulerian Approach

Figures 4 and 5 present instantaneous radial velocity fields (iso-contours) along with the out-of-plan vorticity (iso-lines) and the velocity vectors in different jet regions for the stationary and rotational cases, respectively. The dotted vertical lines correspond to the radial positions discussed in Figure 2.

Figure 4 shows that high radial velocity values are found in the impinging region where a counter-rotating vorticity is generated at the wall due to the vortex ring impingement (Figure 4a). The high radial velocity is observed further radially until the location where the flow separates from the impinging wall just upstream from $r/D = 1.70$. Other flow separation regions from the impinging disk are observed in the outer impinging jet region.

Figure 5 shows differences in the flow dynamics for the rotational case as compared to the stationary case. It is interesting to observe that in the impinging region, the higher radial velocity region is observed farther from the impinging wall (in the vertical direction) as compared to the stationary case ($0.1 < Y/D < 0.2$). It is also shown that the large-scale

vortical structures approach the impinging wall at a different impinging angle for the rotational case (Figure 5b). Consequently, higher radial velocities are observed away from the wall for *r/D* = 0.83 and 1.1 (Figure 2) for the stationary case, whereas an opposite trend is observed farther in the radial direction (*r/D* > 1.7). This is due to the detachment of the vortical structures from the impinging wall for the stationary case, whereas these vortices are advected closer to the impinging wall for the rotational case. A deeper understanding of the vortex dynamics and the flow separation is provided in the present paper using a Lagrangian approach illustrated in the next section.

### 3.4. Spatio-Temporal Description of the Vortex Dynamics; Lagrangian Approach

To better describe the flow physics in different jet regions, an in-depth analysis of the velocity fields is needed. The finite time Lyapunov exponent (FTLE), also called direct Lyapunov exponent (DLE), can be used to identify vortices in the flow. This Lagrangian method was successfully implemented in [1] to describe the vortex dynamics and flow separation in a circular impinging jet. The vortex dynamics for both the stationary and rotational cases is given using the instantaneous velocity and DLE fields, presented in Figures 6 and 7.

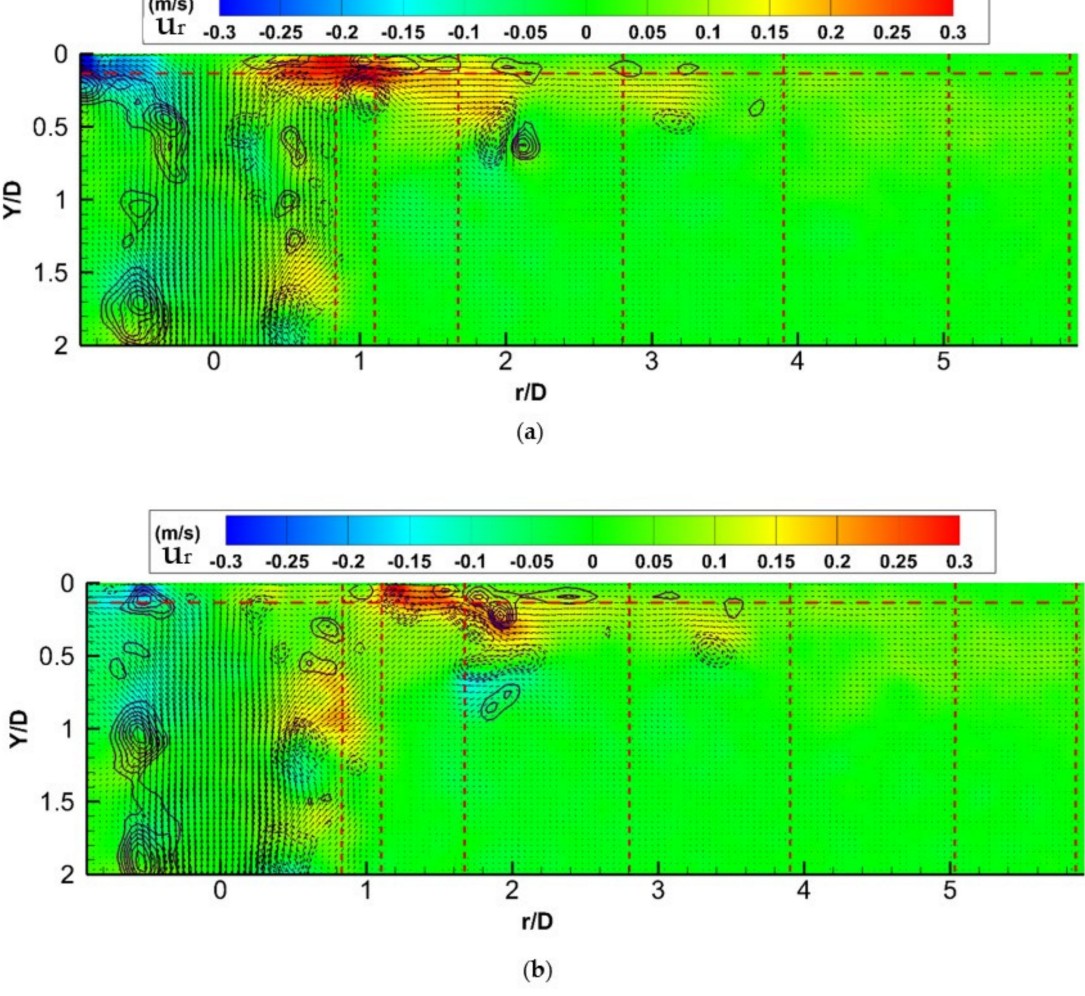

**Figure 4.** Instantaneous velocity fields for the stationary case (**a**) t = t1 and (**b**) t = t1 + 30 ms.

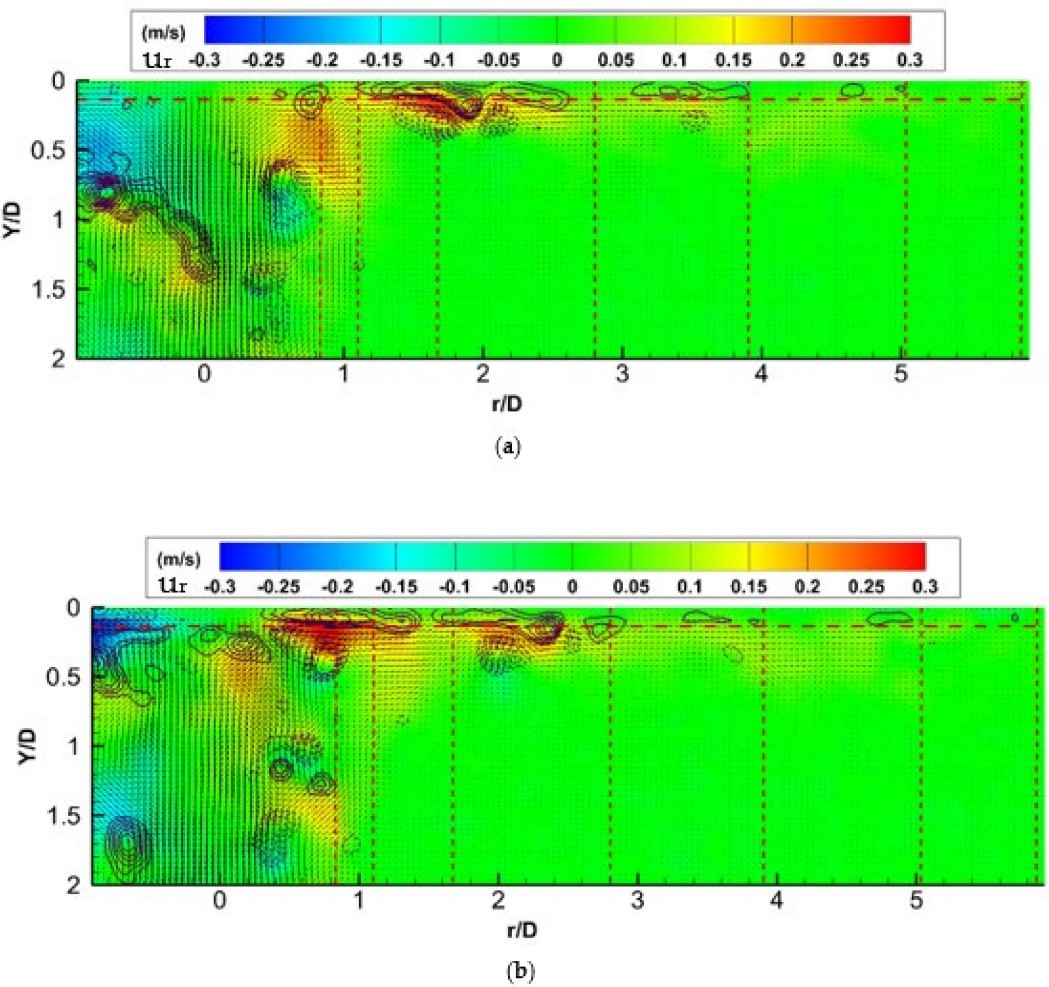

**Figure 5.** Instantaneous velocity fields for the rotational case (**a**) t = t1 and (**b**) t = t1 + 30 ms.

Two representative snapshots of the velocity and DLE fields are shown in Figures 5 and 6 for the stationary and rotational cases, respectively. The time delay between the presented snapshot is 30 ms.

For the stationary case, it is found that the flow separates from the impinging wall just downstream from the jet impact in Region 2 (Figure 6a), whereas the large vortical structures are advected along the impinging wall due to the centrifugal force for the rotating case (Figure 7a). As a consequence, the flow is shown to be attached to the impinging wall when traveling farther radially towards Region 3 due to the centrifugal force (Figure 7a,b). More organized flow structures in the stationary case as compared to the rotational case before the impact of the large-scale vortices in Region 1 are also noted. Such differences would be related to a swirling effect resulting from the centrifugal force and the generation of streamwise vortices in Region 1 of the rotational case. Such vortex dynamics thus confirms the analysis presented from the turbulence intensity fields regarding the effect of centrifugal force on the jet flow dynamics. One can expect a lower wall shear stress amplitude in the rotational case due to the strong flow separation observed with the stationary case. Such vortex dynamics and the interaction between successive separated LCSs influences not only the entrainment of the surrounding fluid but also the ejection/sweep process near the wall [1].

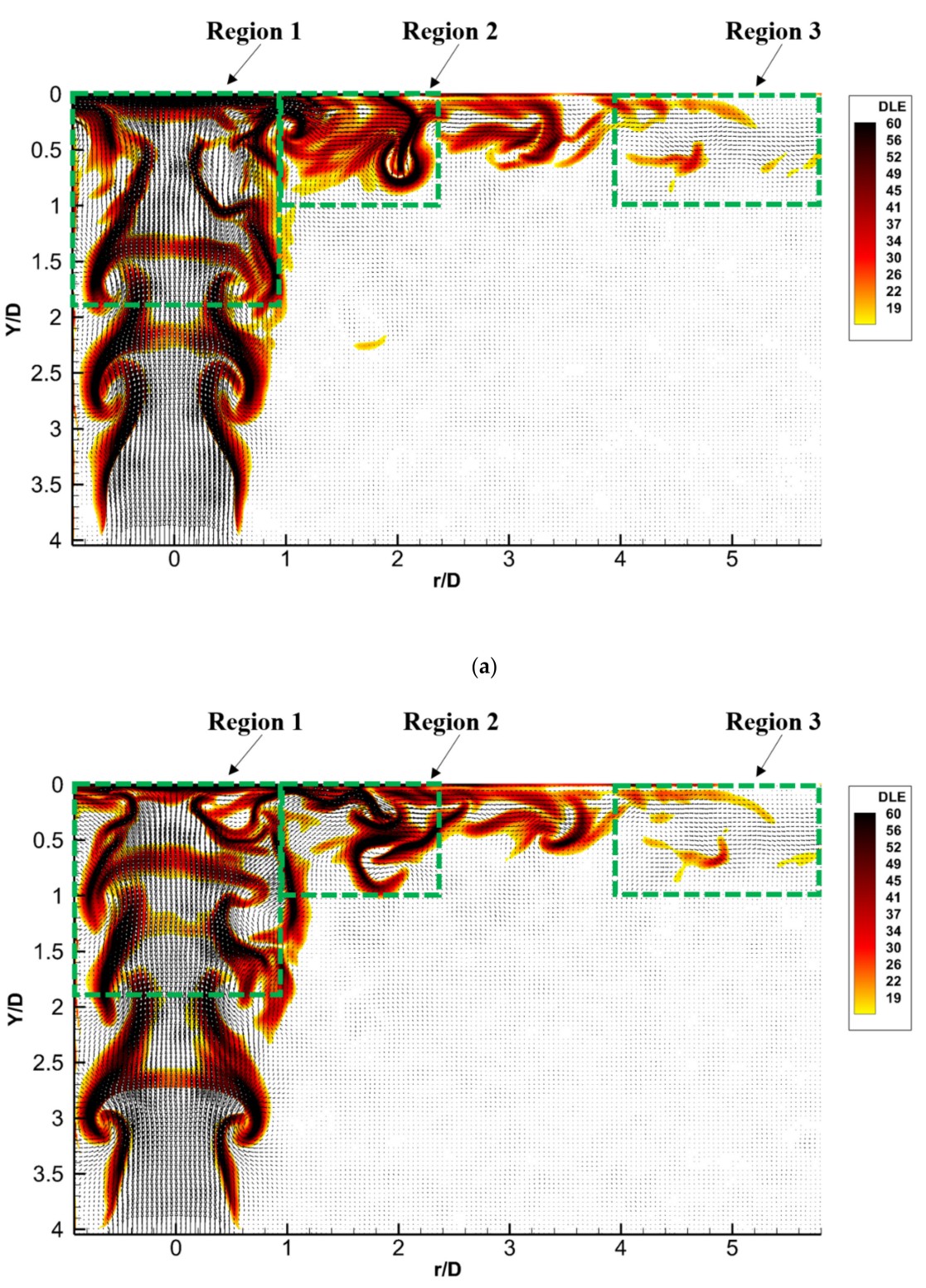

(**a**)

(**b**)

**Figure 6.** FTLE and velocity fields for the stationary case (**a**) t = t1 and (**b**) t = t1 + 30 ms.

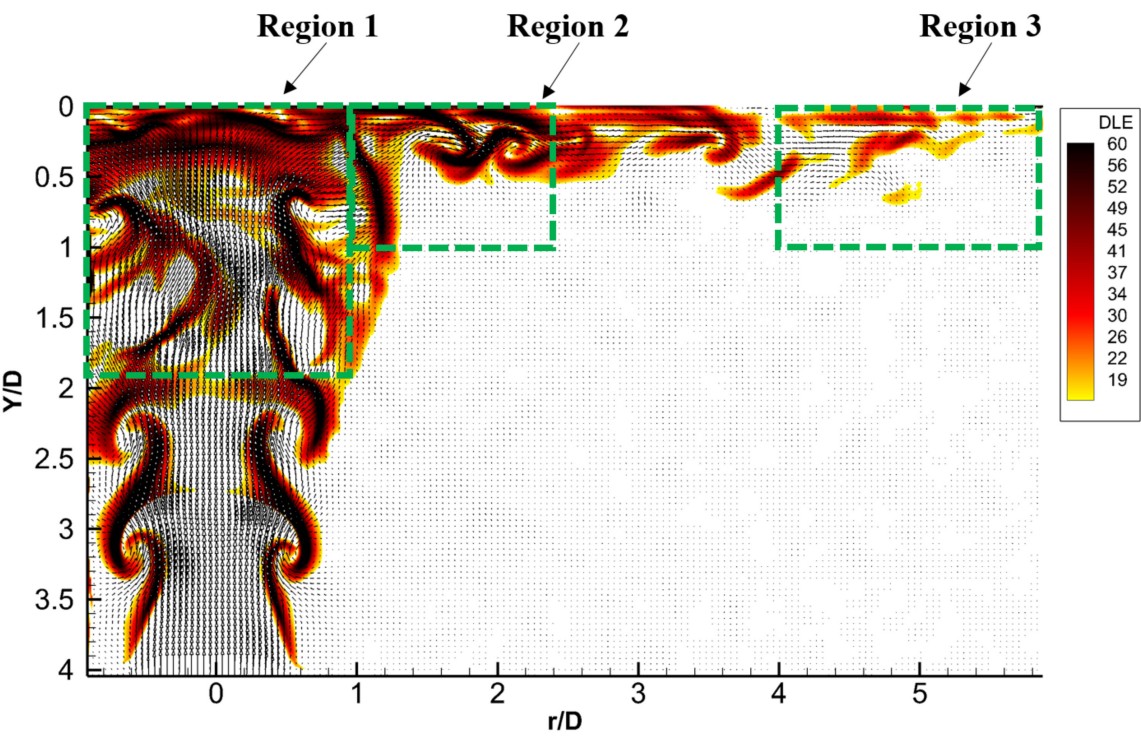

(**a**)

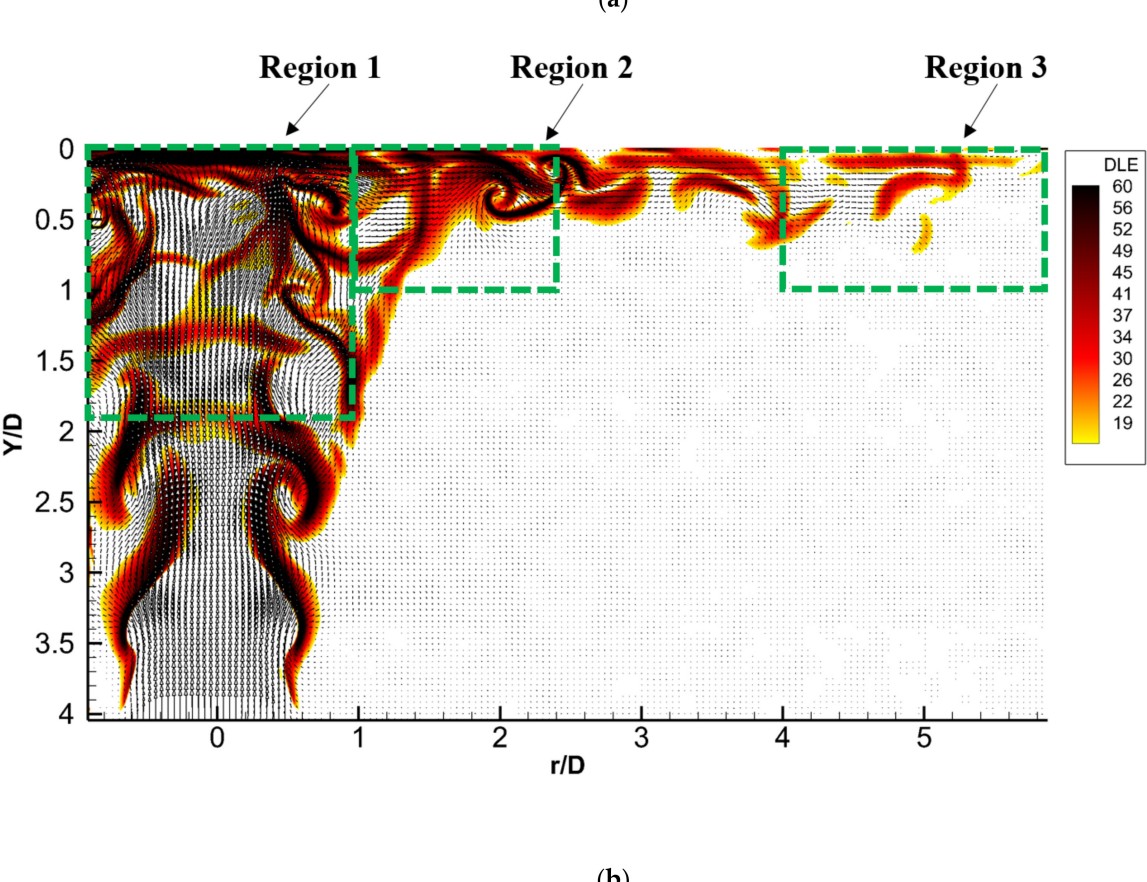

(**b**)

**Figure 7.** FTLE and velocity fields for the rotational case (**a**) t = t1 and (**b**) t = t1 + 30 ms.

## 4. Conclusions

The present experimental investigation illustrates the influence of the centrifugal force of a rotational impinging wall on the vortex dynamics of different regions of a circular jet impinging on a flat wall. It is found that the velocity distribution is affected by the rotating motion in the free jet region, the impinging location and the outer region of the impinging wall. Despite numerous studies on this subject, the influence of the rotating motion on the velocity field was neglected in the impinging zone. It is interesting to note that the inertial force results in a swirling-like dynamics of the large-scale vortices just upstream from their impact on the rotating disk. For the stationary case, the flow separation would be responsible for the lower turbulence intensity and higher radial velocity observed in the near-wall impinging region. It is also found that while the boundary layer detaches from the wall just after the vortex rings impact, the large-scale vortices are advected farther radially along the impinging wall for the rotational case. The present findings are of high interest for both the scientific and engineering communities. For example, a better understanding of the flow physics in impinging jets helps to improve the design of engineering devices that requires enhanced heat transfer or particular wall friction distributions.

The present preliminary investigation illustrates the new flow physics findings of jets impinging on both stationary and rotational disks. Further 3D investigation would be of interest to better describe the nature of different vortical structures and their interactions.

**Author Contributions:** M.E.H.—Data curation, Formal analysis, Methodology, Writing—original draft. D.S.N.—Funding acquisition, Project administration, Writing—review and editing. All authors have read and agreed to the published version of the manuscript.

**Funding:** The authors gratefully acknowledge financial support from Natural Sciences and Engineering Research Council (NSERC) of Canada, the Alberta Ingenuity Fund and the Canadian Foundation for Innovation (CFI).

**Institutional Review Board Statement:** Not applicable.

**Informed Consent Statement:** Not applicable.

**Data Availability Statement:** Not applicable.

**Conflicts of Interest:** The authors declare no conflict of interest. The funders had no role in the design of the study; in the collection, analyses or interpretation of data; in the writing of the manuscript or in the decision to publish the results.

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
