# Peer review of "Experimental Investigation of the Vortex Dynamics in Circular Jet Impinging on Rotating Disk"

_fluids, doi:10.3390/fluids7070223_

Round 1
Reviewer 1 Report
The list of my objections is below:
1. What type of impinging jet will be investigated? Continous, synthetic jet? This information does not appear in the introduction, but it should.
2. The literature review is weak and needs to be improved. I propose to describe the different types of impinging jet, for example, as in the article https://doi.org/10.3390/app12094349
3. The scope and design of the experiment should be better discussed.
4. The results presented in the papers are very interesting and well presented. It is a basic investigation but the scientific level is adequate to the level of the journal. It is a pity that the PIV studies were carried out for only one plane. However, the literature is outdated and only one article cited is less than 5 years old. It must necessarily be corrected. Please add at least 5 more articles published after 2018.
Reviewer 2 Report
The vortex dynamics in circular jet impinging on rotating disk is undoubtedly a current topic not only in basic research, but also in a number of technical applications. The authors choose their own research methodology, designed test equipment for experimental research in the selected area of parameters. After studying the article, it can be stated that it contains new interesting findings in this area. Therefore, I do not have any serious comments in terms of the substantive content of the article. The experimental method used and the method of evaluation of measurements are clearly described, the results are presented clearly.
In the introductory part of the article, the reader would welcome a more detailed analysis of published works on this issue, e.g. only in the ScienceDirect database can you find several newer articles on this topic.
In terms of formal processing of the article, it is necessary to rework it in accordance with the template of Fluids journal.
Round 2
Reviewer 1 Report
In its present form, the article can be accepted for publication.
Reviewer 2 Report
The article was modified according to the reviewers' recommendations, which led to its significant improvement.
Abbreviations are used in the article, the meaning of which is obvious from the context, but I believe that the addition of a list of abbreviations at the end of the article would be welcomed by readers.